# Distributed $k$-Clustering for Data with Heavy Noise

**Xiangyu Guo**
University at Buffalo
Buffalo, NY 14260
xiangyug@buffalo.edu

**Shi Li**
University at Buffalo
Buffalo, NY 14260
shil@buffalo.edu

## Abstract

In this paper, we consider the $k$-center/median/means clustering with outliers problems (or the $(k, z)$-center/median/means problems) in the distributed setting. Most previous distributed algorithms have their communication costs linearly depending on $z$, the number of outliers. Recently Guha et al. [10] overcame this dependence issue by considering bi-criteria approximation algorithms that output solutions with $2z$ outliers. For the case where $z$ is large, the extra $z$ outliers discarded by the algorithms might be too large, considering that the data gathering process might be costly. In this paper, we improve the number of outliers to the best possible $(1 + \epsilon)z$, while maintaining the $O(1)$-approximation ratio and independence of communication cost on $z$. The problems we consider include the $(k, z)$-center problem, and $(k, z)$-median/means problems in Euclidean metrics. Implementation of the our algorithm for $(k, z)$-center shows that it outperforms many previous algorithms, both in terms of the communication cost and quality of the output solution.

## 1  Introduction

Clustering is a fundamental problem in unsupervised learning and data analytics. In many real-life datasets, noises and errors unavoidably exist. It is known that even a few noisy data points can significantly influence the quality of the clustering results. To address this issue, previous work has considered the clustering with outliers problem, where we are given a number $z$ on the number of outliers, and need to find the optimum clustering where we are allowed to discard $z$ points, under some popular clustering objective such as $k$-center, $k$-median and $k$-means.

Due to the increase in volumes of real-life datasets, and the emergence of modern parallel computation frameworks such as MapReduce and Hadoop, computing a clustering (with or without outliers) in the distributed setting has attracted a lot of attention in recent years. The set of points are partitioned into $m$ parts that are stored on $m$ different machines, who collectively need to compute a good clustering by sending messages to each other. Often, the time to compute a good solution is dominated by the communications among machines. Many recent papers on distributed clustering have focused on designing $O(1)$-approximation algorithms with small communication cost [2, 13, 10].

Most previous algorithms for clustering with outliers have the communication costs linearly depending on $z$, the number of outliers. Such an algorithm performs poorly when data is very noisy. Consider the scenario where distributed sensory data are collected by a crowd of people equipped with portable sensory devices. Due to different skill levels of individuals and the quality of devices, it is reasonable to assume that a small constant fraction of the data points are unreliable.

Recently, Guha et al. [10] overcame the linear dependence issue, by giving distributed $O(1)$-approximation algorithms for $k$-center/median/means with outliers problems with communication cost independent of $z$. However, the solutions produced by their algorithms have $2z$ outliers. Such a solution discards $z$ more points compared to the (unknown) optimum one, which may greatly

decrease the efficiency of data usage. Consider an example where a research needs to be conducted using the inliers of a dataset containing 10% noisy points; a filtering process is needed to remove the outliers. A solution with $2z$ outliers will only preserve 80% of data points, as opposed to the promised 90%. As a result, the quality of the research result may be reduced.

Unfortunately, a simple example (described in the supplementary material) shows that if we need to produce any multiplicatively approximate solution with only $z$ outliers, then the linear dependence on $z$ can not be avoided. We show that, even deciding whether the optimum clustering with $z$ outliers has cost 0 or not, for a dataset distributed on 2 machines, requires a communication cost of $\Omega(z)$ bits. Given such a negative result and the positive results of Guha et al. [10], the following question is interesting from both the practical and theoretical points of view:

*Can we obtain distributed $O(1)$-approximation algorithms for $k$-center/median/means with outliers that have communication cost independent of $z$ and output solutions with $(1 + \epsilon)z$ outliers, for any $\epsilon > 0$?*

On the practical side, an algorithm discarding $\epsilon z$ additional outliers is acceptable, as the number can be made arbitrarily small, compared to both the promised number $z$ of outliers and the number $n - z$ of inliers. On the theoretical side, the $(1 + \epsilon)$-factor for the number of outliers is the best we can hope for if we are aiming at an $O(1)$-approximation algorithm with communication complexity independent of $z$; thus answering the question in the affirmative can give the tight tradeoff between the number of outliers and the communication cost in terms of $z$.

In this paper, we make progress in answering the above question for many cases. For the $k$-center objective, we solve the problem completely by giving a $(24(1 + \epsilon), 1 + \epsilon)$-bicriteria approximation algorithm with communication cost $O\left(\frac{km}{\epsilon} \cdot \frac{\log \Delta}{\epsilon}\right)$, where $\Delta$ is the aspect ratio of the metric. ($24(1 + \epsilon)$ is the approximation ratio, $1 + \epsilon$ is the multiplicative factor for the number of outliers our algorithm produces; the formal definition appears later.) For $k$-median/means objective, we give a distributed $(1 + \epsilon, 1 + \epsilon)$-bicrteria approximation algorithm for the case of Euclidean metrics. The communication complexity of the algorithm is poly $\left(\frac{1}{\epsilon}, k, D, m, \log \Delta\right)$, where $D$ is the dimension of the underlying Euclidean metric. (The exact communication complexity is given in Theorem 1.2.) Using dimension reduction techniques, we can assume $D = O(\frac{\log n}{\epsilon^2})$, by incurring a $(1+\epsilon)$-distortion in pairwise distances. So, the setting indeed covers a broad range of applications, given that the term "$k$-means clustering" is defined and studied exclusively in the context of Euclidean metrics. The $(1 + \epsilon, 1 + \epsilon)$-bicriteria approximation ratio comes with a caveat: our algorithm has running time exponential in many parameters such as $\frac{1}{\epsilon}, k, D$ and $m$ (though it has no exponential dependence on $n$ or $z$).

## 1.1 Formulation of Problems

We call the $k$-center (resp. $k$-median and $k$-means) problem with $z$ outliers as the $(k, z)$-center (resp. $(k, z)$-median and $(k, z)$-means) problem. Formally, we are given a set $P$ of $n$ points that reside in a metric space $d$, two integers $k \geq 1$ and $z \in [0, n]$. The goal of the problem is to find a set $C$ of $k$ centers and a set $P' \subseteq P$ of $n - z$ points so as to minimize $\max_{p \in P'} d(p, C)$ (resp. $\sum_{p \in P'} d(p, C)$ and $\sum_{p \in P'} d^2(p, C)$), where $d(p, C) = \min_{c \in C} d(p, c)$ is the minimum distance from $p$ to a center in $C$. For all the 3 objectives, given a set $C \subseteq P$ of $k$ centers, the best set $P'$ can be derived from $P$ by removing the $z$ points $p \in P$ with the largest $d(p, C)$. Thus, we shall only use a set $C$ of $k$ centers to denote a solution to a $(k, z)$-center/median/means instance. The *cost* of a solution $C$ is defined as $\max_{p \in P'} d(p, C), \sum_{p \in P'} d(p, C)$ and $\sum_{p \in P'} d^2(p, C)$ respectively for a $(k, z)$-center, median and means instance, where $P'$ is obtained by applying the optimum strategy. The $n - z$ points in $P'$ and the $z$ points in $P \setminus P'$ are called *inliers* and *outliers* respectively in the solution.

As is typical in the machine learning literature, we consider general metrics for $(k, z)$-center, and Euclidean metrics for $(k, z)$-median/means. In the $(k, z)$-center problem, we assume that each point $p$ in the metric space $d$ can be described using $O(1)$ words, and given the descriptions of two points $p$ and $q$, one can compute $d(p, q)$ in $O(1)$ time. In this case, the set $C$ of centers must be from $P$ since these are all the points we have. For $(k, z)$-median/means problem, points in $P$ and centers $C$ are from Euclidean space $\mathbb{R}^D$, and it is not required that $C \subseteq P$. One should treat $D$ as a small number, since dimension reduction techniques can be applied to project points to a lower-dimension space.

**Bi-Criteria Approximation** We say an algorithm for the $(k,z)$-center/median/means problem achieves a bi-criteria approximation ratio (or simply approximation ratio) of $(\alpha, \beta)$, for some $\alpha, \beta \geq 1$, if it outputs a solution with at most $\beta z$ outliers, whose cost is at most $\alpha$ times the cost of the optimum solution with $z$ outliers.

**Distributed Clustering** In the distributed setting, the dataset $P$ is split among $m$ machines, where $P_i$ is the set of data points stored on machine $i$. We use $n_i$ to denote $|P_i|$. Following the communication model of [8] and [10], we assume there is a central coordinator, and communications can only happen between the coordinator and the $m$ machines. The communication cost is measured in the total number of words sent. Communications happen in rounds, where in each round, messages are sent between the coordinator and the $m$ machines. A message sent by a party (either the coordinator or some machine) in a round can only depends on the input data given to the party, and the messages received by the party in previous rounds. As is common in most of the previous results, we require the number of rounds used to be small, preferably a small constant.

Our distributed algorithm needs to output a set $C$ of $k$ centers, as well as an upper bound $L$ on the maximum radius of the generated clusters. For simplicity, only the coordinator needs to know $C$ and $L$. We do not require the coordinator to output the set of outliers since otherwise the communication cost is forced to be at least $z$. In a typical clustering task, each machine $i$ can figure out the set of outliers in its own dataset $P_i$ based on $C$ and $L$ (1 extra round may be needed for the coordinator to send $C$ and $L$ to all machines).

## 1.2 Prior Work

In the centralized setting, we know the best possible approximation ratios of 2 and 3 [4] for the $k$-center and $(k,z)$-center problems respectively, and thus our understanding in this setting is complete. There has been a long stream of research on approximation algorithms $k$-median and $k$-means, leading to the current best approximation ratio of 2.675 [3] for $k$-median, 9 [1] for $k$-means, and 6.357 for Euclidean $k$-means [1]. The first $O(1)$-approximation algorithm for $(k,z)$-median is given by Chen, [7]. Recently, Krishnaswamy et al. [12] developed a general framework that gives $O(1)$-approximations for both $(k,z)$-median and $(k,z)$-means.

Much of the recent work has focused on solving $k$-center/median/means and $(k,z)$-center/median/means problems in the distributed setting [9, 2, 11, 13, 11, 13, 8, 6, 10, 5]. Many distributed $O(1)$ approximation algorithms with small communication complexity are known for these problems. However, for $(k,z)$-center/median/means problems, most known algorithms have communication complexity linearly depending on $z$, the number of outliers. Guha et al. [10] overcame the dependence issue, by giving $(O(1), 2+\epsilon)$-bicriteria approximation algorithms for all the three objectives. The communication costs of their algorithms are $\tilde{O}(m/\epsilon + mk)$, where $\tilde{O}$ hides a logarithmic factor.

## 1.3 Our Contributions

Our main contributions are in designing $(O(1), 1+\epsilon)$-bicriteria approximation algorithms for the $(k,z)$-center/median/means problems. The algorithm for $(k,z)$-center works for general metrics:

**Theorem 1.1.** *There is a 4-round, distributed algorithm for the $(k,z)$-center problem, that achieves a $(24(1+\epsilon), 1+\epsilon)$-bicriteria approximation and $O\left(\frac{km}{\epsilon} \cdot \frac{\log \Delta}{\epsilon}\right)$ communication cost, where $\Delta$ is the aspect ratio of the metric.*

We give a high-level picture of the algorithm. By guessing, we assume that we know the optimum cost $L^*$ (since we do not know, we need to lose the $\frac{\log \Delta}{\epsilon}$-factor in the communication complexity). In the first round of the algorithm, each machine $i$ will call a procedure called aggregating, on its set $P_i$. This procedure performs two operations. First, it discards some points from $P_i$; second, it moves each of the survived points by a distance of at most $O(1)L^*$. After the two operations, the points will be *aggregated* at a few locations. Thus, machine $i$ can send a compact representation of these points to the coordinator: a list of $(p, w'_p)$ pairs, where $p$ is a location and $w'_p$ is the number of points aggregated at $p$. The coordinator will collect all the data points from all the machines, and run the algorithm of [4] for $(k,z')$-center instance on the collected points, for some suitable $z'$.

To analyze the algorithm, we show that the set $P'$ of points collected by the coordinator well-approximates the original set $P$. The main lemma is that the total number of non-outliers removed by the aggregation procedure on all machines is at most $\epsilon z$. This incurs the additive factor of $\epsilon z$ in the number of outliers. We prove this by showing that inside any ball of radius $L^*$, and for every machine $i \in [m]$, we removed at most $\frac{\epsilon z}{km}$ points in $P_i$. Since the non-outliers are contained in the union of $k$ balls of radius $L^*$, and there are $m$ machines, the total number of removed non-outliers is at most $\epsilon z$. For each remaining point, we shift it by a distance of $O(1)L^*$, leading to an $O(1)$-loss in the approximation ratio of our algorithm.

We perform experiments comparing our main algorithm stated in Theorem 1.1 with many previous ones on real-world datasets. The results show that it matches the state-of-art method in both solution quality (objective value) and communication cost. We remark that the qualities of solutions are measured w.r.t removing only $z$ outliers. Theoretically, we need $(1 + \epsilon)z$ outliers in order to achieve an $O(1)$-approximation ratio and our constant 24 is big. In spite of this, empirical evaluations suggest that the algorithm on real-word datasets performs much better than what can be proved theoretically in the worst case.

For $(k, z)$-median/means problems, our algorithm works for the Euclidean metric case and has communication cost depending on the dimension $D$ of the Euclidean space. One can w.l.o.g. assume $D = O(\log n / \epsilon^2)$ by using the dimension reduction technique. Our algorithm is given in the following theorem:

**Theorem 1.2.** *There is a 2-round, distributed algorithm for the $(k, z)$-median/means problems in D-dimensional Euclidean space, that achieves a $(1 + \epsilon, 1 + \epsilon)$-bicriteria approximation ratio with probability $1 - \delta$. The algorithm has communication cost $O\left(\Phi D \cdot \frac{\log(n\Delta/\epsilon)}{\epsilon}\right)$, where $\Delta$ is the aspect ratio of the input points, $\Phi = O\left(\frac{1}{\epsilon^2}(kD + \log\frac{1}{\delta}) + mk\right)$ for $(k, z)$-median, and $\Phi = O\left(\frac{1}{\epsilon^4}(kD + \log\frac{1}{\delta}) + mk\log\frac{mk}{\delta}\right)$ for $(k, z)$-means.*

We now give an overview of our algorithm for $(k, z)$-median/means. First, it is not hard to reformulate the objective of the $(k, z)$-median problem as minimizing $\sup_{L \geq 0} \left(\sum_{p \in P} d_L(p, C) - zL\right)$, where $d_L$ is obtained from $d$ by truncating all distances at $L$. By discretization, we can construct a set $\mathbb{L}$ of $O\left(\frac{\log(\Delta n/\epsilon)}{\epsilon}\right)$ interesting values that the $L$ under the superior operator can take. Thus, our goal becomes to find a set $C$, that is simultaneously good for every $k$-median instance defined by $d_L, L \in \mathbb{L}$. Since now we are handling $k$-median instances (without outliers), we can use the communication-efficient algorithm of [2] to construct an $\epsilon$-coreset $Q_L$ with weights $w_L$ for every $L \in \mathbb{L}$. Roughly speaking, the coreset $Q_L$ is similar to the set $P$ for the task of solving the $k$-median problem under metric $d_L$. The size of each $\epsilon$-coreset $Q_L$ is at most $\Phi$, implying the communication cost stated in the theorem. After collecting all the coresets, the coordinator can approximately solve the optimization problem on them. This will lead to an $(1 + O(\epsilon), 1 + O(\epsilon))$-bicriteria approximate solution. The running time of the algorithm, however, is exponential in the total size of the coresets. The argument can be easily adapted to the $(k, z)$-means setting.

**Organization**   In Section 2, we prove Theorem 1.1, by giving the $(24(1 + \epsilon), 1 + \epsilon)$-approximation algorithm. The empirical evaluations of our algorithm for $(k, z)$-center and the proof of Theorem 1.2 are provided in the supplementary material.

**Notations**   Throughout the paper, point sets are multi-sets, where each element has its own identity. By a copy of some point $p$, we mean a point with the same description as $p$ but a different identity. For a set $Q$ of points, a point $p$, and a radius $r \geq 0$, we define $\text{ball}_Q(p, r) = \{q \in Q : d(p, q) \leq r\}$ to be the set of points in $Q$ that have distances at most $r$ to $p$. For a weight vector $w \in \mathbb{Z}_{\geq 0}^Q$ on some set $Q$ of points, and a set $S \subseteq Q$, we use $w(S) = \sum_{p \in S} w_p$ to denote the total weight of points in $S$.

Throughout the paper, $P$ is always the set of input points. We shall use $d_{\min} = \min_{p,q \in P: d(p,q) > 0} d(p, q)$ and $d_{\max} = \max_{p,q \in P} d(p, q)$ to denote the minimum and maximum non-zero pairwise distance between points in $P$. Let $\Delta = \frac{d_{\max}}{d_{\min}}$ denote the aspect ratio of the metric.

# 2 Distributed $(k, z)$-Center Algorithm with $(1 + \epsilon)z$ Outliers

In this section, we prove Theorem 1.1, by giving the $(24(1 + \epsilon), 1 + \epsilon)$-approximation algorithm for $(k, z)$-center, with communication cost $O\left(\frac{km}{\epsilon} \cdot \frac{\log \Delta}{\epsilon}\right)$. Let $L^*$ be the cost of the optimum $(k, z)$-center solution (which is not given to us). We assume we are given a parameter $L \geq 0$ and our goal is to design a main algorithm with communication cost $O\left(\frac{km}{\epsilon}\right)$, that either returns a $(k, (1+\epsilon)z)$-center solution of cost at most $24L$, or certifies that $L^* > L$. Notice that $L^* \in \{0\} \cup [d_{\min}/2, d_{\max}]$. We can obtain our $(24(1 + \epsilon), 1 + \epsilon)$-approximation by running the main algorithm for $O\left(\frac{\log \Delta}{\epsilon}\right)$ different values of $L$ in parallel, and among all generated solutions, returning the one correspondent to the smallest $L$. A naive implementation requires all the parties to know $d_{\min}$ and $d_{\max}$ in advance; we show in the supplementary material that the requirement can be removed.

In intermediate steps, we may deal with $(k, z)$-center instances where points have integer weights. In this case, the instance is defined as $(Q, w)$, where $Q$ is a set of points, $w \in \mathbb{Z}_{>0}^Q$, and $z$ is an integer between 0 and $w(Q) = \sum_{q \in Q} w_q$. The instance is equivalent to the instance $\hat{Q}$, the multi-set where we have $w_q$ copies of each $q \in Q$.

[4] gave a 3-approximation algorithm for the $(k, z)$-center problem. However, our setting is slightly more general so we can not apply the result directly. We are given a weighted set $Q$ of points that defines the $(k, z)$-center instance. The optimum set $C^*$ of centers, however, can be from the superset $P \supseteq Q$ which is hidden to us. Thus, our algorithm needs output a set $C$ of $k$ centers from $Q$ and compare it against the optimum set $C^*$ of centers from $P$. Notice that by losing a factor of 2, we can assume centers are in $Q$; this will lead to a 6-approximation. Indeed, by applying the framework of [4] more carefully, we can obtain a 4-approximation for this general setting. We state the result in the following theorem:

**Theorem 2.1** ([4])**.** *Let $d$ be a metric over the set $P$ of points, $Q \subseteq P$ and $w \in \mathbb{Z}_{>0}^Q$. There is an algorithm* kzc *(Algorithm 1) that takes inputs $k, z' \geq 1$, $(Q, w')$ with $|Q| = n'$, the metric $d$ restricted to $Q$, and a real number $L' \geq 0$. In time $O(n'^2)$, the algorithm either outputs a $(k, z')$-center solution $C' \subseteq Q$ to the instance $(Q, w')$ of cost at most $4L'$, or certifies that there is no $(k, z')$-center solution $C^* \subseteq P$ of cost at most $L'$ and outputs "No".*

The main algorithm is dist-kzc (Algorithm 3), which calls an important procedure called aggregating (Algorithm 2). We describe aggregating and dist-kzc in Section 2.1 and 2.2 respectively.

## 2.1 Aggregating Points

The procedure aggregating, as described in Algorithm 2, takes as input the set $Q \subseteq P$ of points to be aggregated (which will be some $P_i$ when we actually call the procedure), the guessed optimum cost $L$, and $y \geq 0$, which controls how many points can be removed from $Q$. It returns a set $Q'$ of points obtained from aggregation, along with their weights $w'$.

| **Algorithm 1** kzc$(k, z', (Q, w'), L')$ | **Algorithm 2** aggregating$(Q, L, y)$ |
|---|---|
| 1: $U \leftarrow Q, C' \leftarrow \emptyset$; | 1: $U \leftarrow Q, Q' \leftarrow \emptyset$; |
| 2: **for** $i \leftarrow 1$ to $k$ **do** | 2: **while** $\exists p \in Q$ with $|\mathsf{ball}_U(p, 2L)| > y$ **do** |
| 3: $\quad p_i \leftarrow p \in Q$ with largest $w'(\mathsf{ball}_U(p, 2L'))$ | 3: $\quad Q' \leftarrow Q' \cup \{p\}, w'_p \leftarrow |\mathsf{ball}_U(p, 4L)|$ |
| 4: $\quad C' \leftarrow C' \cup \{p_i\}$ | 4: $\quad U \leftarrow U \setminus \mathsf{ball}_U(p, 4L)$ |
| 5: $\quad U \leftarrow U \setminus \mathsf{ball}_U(p_i, 4L')$ | 5: **return** $(Q', w')$ |
| 6: **if** $w'(U) > z'$ **then return** "No" **else return** $C'$ | |

In aggregating, we start from $U = Q$ and $Q' = \emptyset$ and keep removing points from $U$. In each iteration, we check if there is a $p \in Q$ with $|\mathsf{ball}_U(p, 2L)| \geq y$. If yes, we add $p$ to $Q'$, remove $\mathsf{ball}_U(p, 4L)$ from $U$ and let $w_p$ be the number of points removed. We repeat thie procedure until such a $p$ can not be found. We remark that the procedure is very similar to the algorithm kzc (Algorithm 1) in [4].

We start from some simple observations about the algorithm.

**Claim 2.2.** *We define $V = \bigcup_{p \in Q'} \mathsf{ball}_Q(p, 4L)$ to be the set of points in $Q$ with distance at most $4L$ to some point in $Q'$ at the end of Algorithm 2. Then, the following statements hold at the end of the algorithm:*

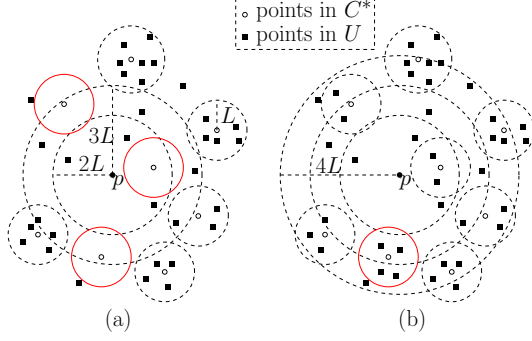
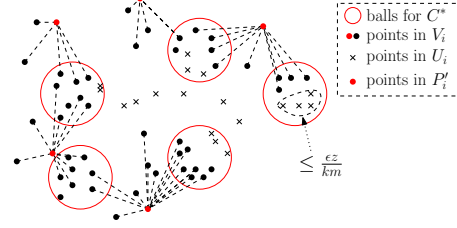

Figure 1: Two cases in proof of Lemma 2.3. In Figure (a), the balls $\{\mathsf{ball}_U(c, L) : c \in C^*, d(p, c) \le 3L\}$ (red circles) are all empty. So, $\mathsf{ball}_U(p, 2L) \subseteq O$. In Figure (b), there is a non-empty $\mathsf{ball}_U(c, L)$ for some $c \in C^*$ with $d(p, c) \le 3L$ (the red circle). The ball is contained in $\mathsf{ball}_U(p, 4L)$.

Figure 2: Illustration for proof of Lemma 2.7. $f_i : V_i \to P'_i$ is indicated by the dashed lines, each of whom is of length at most $4L$. The number of crosses in a circle is at most $\frac{\epsilon z}{km}$.

1. $U = Q \setminus V$.

2. $\big|\mathsf{ball}_U(p, 2L)\big| \le y$ for every $p \in Q$.

3. There is a function $f : V \to Q'$ such that $d(p, f(p)) \le 4L, \forall p \in V$, and $w'(q) = |f^{-1}(q)|, \forall q \in Q'$.

*Proof.* $U$ is exactly the set of points in $Q$ with distance more than $4L$ to any point in $Q'$ and thus $U = Q \setminus V$. Property 2 follows from the termination condition of the algorithm. Property 3 holds by the way we add points to $Q'$ and remove points from $U$. If in some iteration we added $q$ to $Q'$, we can define $f(p) = q$ for every point $p \in \mathsf{ball}_U(p, 4L)$, i.e, every point removed from $U$ in the iteration. $\qquad\square$

We think of $U$ as the set of points we discard from $Q$ and $V$ as the set of survived points. We then move each $p \in V$ to $f(p) \in Q'$ and thus $V$ will be aggregated at the set $Q'$ of locations. The following crucial lemma upper bounds $|Q'|$:

**Lemma 2.3.** *Let $\hat{z} \ge 0$ and assume there is a $(k, \hat{z})$-center solution $C^* \subseteq P$ to the instance $Q$ with cost at most $L$. Then, at the end of Algorithm 2 we have $|Q'| \le k + \frac{\hat{z}}{y}$.*

*Proof.* Let $O = Q \setminus \bigcup_{c \in C^*} \mathsf{ball}_Q(c, L)$ be the set of outliers according to solution $C^*$. Thus $|O| \le \hat{z}$.

Focus on the moment before we run Step 3 in some iteration of aggregating. See Figure 1 for the two cases we are going to consider. In case (a), every center $c \in \mathsf{ball}_{C^*}(p, 3L)$ has $\mathsf{ball}_U(c, L) = \emptyset$. In this case, every point $q \in \mathsf{ball}_U(p, 2L)$ has $d(q, C^*) > L$: if $d(p, c) > 3L$ for some $c \in C^*$, then $d(q, c) \ge d(p, c) - d(p, q) > 3L - 2L = L$ by triangle inequality; for some $c \in C^*$ with $d(p, c) \le 3L$, we have $\mathsf{ball}_U(c, L) = \emptyset$, implying that $d(q, c) > L$ as $q \in U$. Thus, $\mathsf{ball}_U(p, 2L) \subseteq O$. So, Step 3 in this iteration will decrease $|O \cap U|$ by at least $|\mathsf{ball}_U(p, 4L)| \ge |\mathsf{ball}_U(p, 2L)| > y$.

Consider the case (b) where some $c \in \mathsf{ball}_{C^*}(p, 3L)$ has $\mathsf{ball}_U(c, L) \ne \emptyset$. Then $\mathsf{ball}_U(p, 4L) \supseteq \mathsf{ball}_U(c, L)$ will be removed from $U$ by Step 3 in this iteration. Thus,

1. if case (a) happens, then $|U \cap O|$ is decreased by more than $y$ in this iteration;

2. otherwise case (b) happens; then for some $c \in C^*$, $\mathsf{ball}_U(c, L)$ changes from non-empty to $\emptyset$.

The first event can happen for at most $|O|/y \le \hat{z}/y$ iterations and the second event can happen for at most $|C^*| \le k$ iterations. So, $|Q'| \le k + \hat{z}/y$. $\qquad\square$

## 2.2 The Main Algorithm

We are now ready to describe the main algorithm for the $(k, z)$-center problem, given in Algorithm 3. In the first round, each machine will call $\mathsf{aggregating}(P_i, L, \frac{\epsilon z}{km})$ to obtain $(P_i', w_i')$. All the machines will first send their corresponding $|P_i'|$ to the coordinator. In Round 2 the algorithm will check if $\sum_{i \in [m]} |P_i'|$ is small or not. If yes, send a "Yes" message to all machines; otherwise return "No" and terminate the algorithm. In Round 3, if a machine $i$ received a "Yes" message from the coordinator, then it sends the dataset $P_i'$ with the weight vector $w_i'$ to the coordinator. Finally in Round 4, the coordinator collects all the weighted points $P' = \bigcup_{i \in [m]} P_i'$ and run kzc on these points.

---

**Algorithm 3** dist-kzc

---

**input on all parties**: $n, k, z, m, L, \epsilon$
**input on machine** $i$: dataset $P_i$ with $|P_i| = n_i$
**output**: a set $C' \subseteq P$ or "No" (which certifies $L^* > L$)

---

**Round 1 on machine** $i \in [m]$
1: $(P_i', w_i') \leftarrow \mathsf{aggregating}(P_i, L, \frac{\epsilon z}{km})$
2: send $|P_i'|$ to the coordinator

---

**Round 2 on the coordinator**
1: **if** $\sum_{i \in [m]} |P_i'| > km(1 + 1/\epsilon)$ **then return** "No" **else** send "Yes" to each machine $i \in [m]$

---

**Round 3 on machine** $i \in [m]$
1: Upon receiving of a "Yes" message from the coordinator, respond by sending $(P_i', w_i')$

---

**Round 4 on the coordinator**
1: let $P' \leftarrow \bigcup_{i=1}^m P_i'$
2: let $w'$ be the function from $P'$ to $\mathbb{Z}_{>0}$ obtained by merging $w_1', w_2', \cdots, w_m'$
3: let $z' \leftarrow (1 + \epsilon)z + w'(P') - n$
4: **if** $z' < 0$ **then return** "No" **else return** $\mathsf{kzc}(k, z', (P', w'), L' = 5L)$

---

An immediate observation about the algorithm is that its communication cost is small:

**Claim 2.4.** *The communication cost of* dist-kzc *is* $O(\frac{km}{\epsilon})$.

*Proof.* The total communication cost of Round 1 and Round 2 is $O(m)$. We run Round 3 only when the coordinator sent the "Yes" message, in which case the communication cost is at most $\sum_{i=1}^m |P_i'| \le km(1 + 1/\epsilon) = O(\frac{km}{\epsilon})$. $\square$

It is convenient to define some notations before we make further analysis. For every machine $i \in [m]$, let $P_i'$ be the $P_i'$ constructed in Round 1 on machine $i$. Let $V_i = \bigcup_{p \in P_i'} \mathsf{ball}_{P_i}(p, 4L)$ be the set of points in $P_i$ that are within distance at most $4L$ to some point in $P_i'$. Notice that this is the definition of $V$ in Claim 2.2 for the execution of aggregating on machine $i$. Let $U_i = P_i \setminus V_i$; this is the set $U$ at the end of this execution. Let $f_i$ be the mapping from $V_i$ to $P_i'$ satisfying Property 3 of Claim 2.2. Let $V = \bigcup_{i \in [m]} V_i, P' = \bigcup_{i \in [m]} P_i'$ and $f$ be the function from $V$ to $P'$, obtained by merging $f_1, f_2, \cdots, f_m$. Thus $(p, f(p)) \le 4L, \forall p \in V$ and $w'(q) = |f^{-1}(q)|, \forall q \in P'$.

**Claim 2.5.** *If* dist-kzc *returns a set* $C'$, *then* $C'$ *is a* $(k, (1 + \epsilon)z)$-*center solution to the instance* $P$ *with cost at most* $24L$.

*Proof.* $C'$ must be returned in Step 4 in Round 4. By Theorem 2.1 for kzc, $C'$ is a $(k, z')$-center solution to $(P', w')$ of cost at most $4 \cdot 5L = 20L$. That is, $w'\left(P' \setminus \bigcup_{c \in C'} \mathsf{ball}_{P'}(c, 20L)\right) \le z'$. This implies $w'\left(\bigcup_{c \in C'} \mathsf{ball}_{P'}(c, 20L)\right) \ge w'(P') - z' = n - (1 + \epsilon)z$. Notice that for each $q \in P'$, the set $f^{-1}(q) \subseteq V \subseteq P$ of points are within distance $4L$ from $q$ and $w'(q) = |f^{-1}(q)|$. So, $\left|\bigcup_{c \in C'} \mathsf{ball}_P(c, 24L)\right| \ge n - (1 + \epsilon)z$, which is exactly $\left|P \setminus \bigcup_{c \in C'} \mathsf{ball}_P(c, 24L)\right| \le (1 + \epsilon)z$. $\square$

We can now assume $L \ge L^*$ and we need to prove that we must reach Step 4 in Round 4 and return a set $C'$. We define $C^* \subseteq P$ to be a set of size $k$ such that $|P \setminus \bigcup_{c \in C^*} \mathsf{ball}(c, L)| \le z$. Let $I = \bigcup_{c \in C^*} \mathsf{ball}_P(c, L)$ be the set of "inliers" according to $C^*$ and $O = P \setminus I$ be the set of outliers. Thus, $|I| \ge n - z$ and $|O| \le z$.

**Lemma 2.6.** *After Round 1, we have* $\sum_{i \in [m]} |P_i'| \le km(1 + 1/\epsilon)$.

*Proof.* Let $z_i = |P_i \cap O| = \left| P_i \setminus \bigcup_{c \in C^*} \mathsf{ball}_{P_i}(c, L) \right|$ be the set of outliers in $P_i$. Then, $C^*$ is a $(k, z_i)$-center solution to the instance $P_i$ with cost at most $L$. By Lemma 2.3, we have that $|P_i'| \leq k + \frac{z_i}{\epsilon z/(km)}$. So, we have

$$\sum_{i \in [m]} |P_i'| \leq km + \frac{km}{\epsilon z} \sum_{i \in [m]} z_i \leq km \left( 1 + \frac{1}{\epsilon} \right). \qquad \square$$

Therefore, the coordinator will not return "No" in Round 2. It remains to prove the following Lemma.

**Lemma 2.7.** *Algorithm 3 will reach Step 4 in Round 4 and return a set $C'$.*

*Proof.* See Figure 2 for the illustration of the proof. By Property 2 of Claim 2.2, we have $|\mathsf{ball}_{U_i}(p, 2L)| \leq \frac{\epsilon z}{km}$ for every $p \in U_i$ since $U_i \subseteq P_i$. This implies that for every $c \in C^*$, we have $|\mathsf{ball}_{U_i}(c, L)| \leq \frac{\epsilon z}{km}$. (Otherwise, taking an arbitrary $p$ in the ball leads to a contradiction.)

$$|U_i \cap I| = \left| \bigcup_{c \in C^*} \mathsf{ball}_{U_i}(c, L) \right| \leq \sum_{c \in C^*} |\mathsf{ball}_{U_i}(c, L)| \leq \sum_{c \in C^*} \frac{\epsilon z}{km} \leq \frac{\epsilon z}{m}, \quad \forall i \in [m].$$

$$\sum_{i \in [m]} |I \cap V_i| = \sum_{i \in [m]} \left( |I \cap P_i| - |I \cap U_i| \right) \geq \sum_{i \in [m]} \left( |I \cap P_i| - \frac{\epsilon z}{m} \right) = |I| - \epsilon z \geq n - (1 + \epsilon)z.$$

For every $p \in V \cap I$, $f(p)$ will have distance at most $L + 4L = 5L$ to some center in $C^*$. Also, notice that $w'(q) = |f^{-1}(q)|$ for every $q \in P'$, we have that

$$w'\left( \bigcup_{c \in C^*} \mathsf{ball}_{P'}(c, 5L) \right) \geq |V \cap I| \geq n - (1 + \epsilon)z.$$

So, $w'(P' \setminus \bigcup_{c \in C^*} \mathsf{ball}_{P'}(c, 5L)) \leq w(P') - n + (1 + \epsilon)z = z'$. This implies that $z' \geq 0$, and there is a $(k, z')$-center solution $C^* \subseteq P$ to the instance $(P', w')$ of cost at most $5L$. Thus dist-kzc will reach Step 4 in Round 4 and returns a set $C'$. This finishes the proof of the Lemma. $\square$

We now briefly analyze the running times of algorithms on all parties. The running time of computing $P_i'$ on each machine $i$ in round 1 is $O(n_i^2)$ and this is the bottleneck for machine $i$. Considering all possible values of $L$, the running time on machine $i$ is $O\left( n_i^2 \cdot \frac{\log \Delta}{\epsilon} \right)$. The running time of the round-4 algorithm of the central coordinator for one $L$ will be $O\left( \left( \frac{km}{\epsilon} \right)^2 \right)$. We sort all the interesting $L$ values in increasing order. The central coordinator can use binary search to find some $L'$ such that the main algorithm outputs a set $C'$ for $L = L'$ but outputs "No" for $L$ being the value before $L'$ in the ordering. So, the running time of the central coordinator can be made $O\left( \left( \frac{km}{\epsilon} \right)^2 \cdot \log \frac{\log \Delta}{\epsilon} \right)$.

The quadratic dependence of running time of machine $i$ on $n_i$ might be an issue when $n_i$ is big; we discuss how to alleviate the issue in the supplementary material.

## 3  Conclusion

In this paper, we give a distributed $(24(1 + \epsilon), 1 + \epsilon)$-bicriteria approximation for the $(k, z)$-center problem, with communication cost $O\left( \frac{km}{\epsilon} \cdot \frac{\log \Delta}{\epsilon} \right)$. The running times of the algorithms for all parties are polynomial. We evaluate the algorithm on realworld data sets and it outperforms most previous algorithms, matching the performance of the state-of-art method[10].

For the $(k, z)$-median/means problem, we give a distributed $(1 + \epsilon, 1 + \epsilon)$-bicriteria approximation algorithm with communication cost $O\left( \Phi D \cdot \frac{\log \Delta}{\epsilon} \right)$, where $\Phi$ is the upper bound on the size of the coreset constructed using the algorithm of [2]. The central coordinator needs to solve the optimization problem of finding a solution that is simultaneously good for $O\left( \frac{\log(\Delta n/\epsilon)}{\epsilon} \right)$ $k$-median/means instances. Since the approximation ratio for this problem will go to *both* factors in the bicriteria ratio, we really need a $(1 + \epsilon)$-approximation for the optimization problem. Unfortunately, solving $k$-median/means alone is already APX-hard, and we don't know a heuristic algorithm that works well in practice (e.g, a counterpart to Lloyd's algorithm for $k$-means). It is interesting to study if a different approach can lead to a polynomial time distributed algorithm with $O(1)$-approximation guarantee.

**Acknowledgments**

This research was supported by NSF grants CCF-1566356 and CCF-1717134.

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
