[Supplementary Material]

# Distributed $k$-Clustering for Data with Heavy Noise (Supplementary Material)

**Xiangyu Guo**
University at Buffalo
Buffalo, NY 14260
xiangyug@buffalo.edu

**Shi Li**
University at Buffalo
Buffalo, NY 14260
shil@buffalo.edu

## A  Necessity of Linear Dependence of Communication Cost on $z$ for True Approximation Algorithms

In this section, we show that if one is aiming for a multiplicative approximation for the $(k, z)$-center, $(k, z)$-median, or $(k, z)$-means problem, then the communication cost is at least $\Omega(z)$ bits, even if there are only 2 machines. We show that deciding whether the optimum $(k, z)$-center solution has cost 0 or not requires $\Omega(z)$ bits of communication. This holds for any combination of values for $n, k$ and $z$ such that $k + z \leq n - 1$. Let $B = 1$. The points are all in the real line $\mathbb{R}$. On machine 1, there are $n - z - 2$ copies of points from the set $\{-1, -2, \cdots, -(k-1)\}$, where each one of the $k - 1$ points appears either $\left\lfloor \frac{n-z-2}{k-1} \right\rfloor$ or $\left\lceil \frac{n-z-2}{k-1} \right\rceil$ times. Notice that each point in the set appears at least once in the set. Meanwhile, machine 1 has a set $A$ of different points in $[2(z+2)]$, and machine 2 has a set $B$ of different points in $[2(z+2)]$, and we have $|A| + |B| = z + 2$. If $A \cap B \neq \emptyset$, then the cost of the optimum solution is 0. Let $e \in A \cap B$, then we can discard all points except $e$ from $A$ and $B$. Then we discarded exactly $z$ points and the remaining set of points are at $k - 1 + 1 = k$ locations. On the other hand, if $A \cap B = \emptyset$, then the cost of the optimum solution is not 0. Thus deciding whether the cost is 0 or not requires us to decide if $A \cap B = \emptyset$, which is exactly the *set disjointness* problem. This requires a communication cost of $\Omega(z)$ between machine 1 and machine 2[1].

## B  Dealing with Various Issues of the Algorithm for $(k, z)$-Center

In this section, we show how to handle various issues that our $(k, z)$-center algorithm might face.

**When $d_{\min}$ and $d_{\max}$ are not given.** We can remove the assumption that $d_{\min}$ and $d_{\max}$ are given to us. Let $d_{\min,i}$ and $d_{\max,i}$ be the minimum and maximum non-zero pairwise distances between points in $P_i$. The crucial observation is that running aggregating on $P_i$ for $L < d_{\min,i}$ is the same as running it for $L = 0$, and running it for $L > d_{\max,i}$ is the same as running it for $L = d_{\max,i}$. Thus, machine $i$ only needs to consider $L$ values that are integer powers of $1+\epsilon$ inside $[d_{\min,i}, (1+\epsilon)d_{\max,i})$, or 0, and send results for these $L$ values. Since $d_{\min,i} \geq d_{\min}$ and $d_{\max,i} \geq l_{\max}$, the number of such $L$ values is at most $O\left(\frac{\log \Delta}{\epsilon}\right)$. Also notice that the data points sent from machine $i$ to the coordinator are all generated from $P_i$. Thus, the aspect ratio for the union of all points received by the coordinator, is at most $\Delta$. This can guarantee that the coordinator only needs to use $O(\log \frac{\log \Delta}{\epsilon})$ iterations in the binary search step in Round 4.

**When $\Delta$ is super big.** There are many ways to handle the case when $\Delta$ is super-large. In many applications, we know the nature of the dataset and have a reasonable guess on $L^*$. In other applications, we may be only interested in the case where $L^* \in [A, B]$: we are happy with any clustering of cost less than $A$ and any clustering of cost more than $B$ is meaningless. In these

applications where we have inside information about the dataset, the number of guesses can be much smaller. Finally, if we allow more rounds in our algorithm, we can use binary search for the whole algorithm dist-kzc, not just inside Round 4. We only need to run the algorithm for $O\left(\log \frac{\log \Delta}{\epsilon}\right)$ iterations; this will increase the number of rounds to $O\left(\log \frac{\log \Delta}{\epsilon}\right)$.

**Handling the Quadratic Running Time of Round 1 on Machine $i$.** In Round 1 of the algorithm dist-kzc, each machine $i$ needs to run aggregating on $n_i = |P_i|$ points, leading to a running time of order $O(n_i^2)$. In cases where $n_i$ is large, the algorithm might be slow. We can decrease the running time, at the price of increasing the communication cost and the running time on the coordinator. We view each $i \in [m]$ as a collection of $t_i \geq 1$ sub-machines, for some integer $t_i \in [1, n_i]$. Then, we run dist-kzc on the set of $\sum_{i \in [m]} t_i$ sub-machines, instead of the original set of $m$ machines. The communication cost of the algorithm dist-kzc increases to $O\left(\frac{k \sum_{i \in [m]} t_i}{\epsilon} \cdot \frac{\log \Delta}{\epsilon}\right)$, and the running time on each machine $i$ decreases to $O\left(\left(\frac{n_i}{t_i}\right)^2 \cdot t_i \cdot \frac{\log \Delta}{\epsilon}\right) = O\left(\frac{n_i^2}{t_i} \cdot \frac{\log \Delta}{\epsilon}\right)$, and the running time of the algorithm for the coordinator becomes $O\left(\left(\frac{k \sum_{i \in [m]} t_i}{\epsilon}\right)^2 \cdot \log \frac{\log \Delta}{\epsilon}\right)$. Each machine $i$ can choose a $t_i$ so that the $O\left(\frac{n_i^2}{t_i} \cdot \frac{\log \Delta}{\epsilon}\right)$-time algorithm of Round 1 terminates in acceptable amount of time.

## C  Distributed Algorithms $(k, z)$-Median/Means

In this section, we give our distributed algorithm for the $(k, z)$-median/means problems in Euclidean metrics. Let $m, k, z, \epsilon, n, P \subseteq \mathbb{R}^D$ and $\{P_i\}_{i \in [m]}$ be as defined in the problem setting. Let $\delta > 0$ be the confidence parameter; i.e, our algorithm needs to succeed with probability $1 - \delta$. Also, we define a parameter $\ell \in \{1, 2\}$ to indicate whether the problem we are considering is $(k, z)$-median ($\ell = 1$) or $(k, z)$-means ($\ell = 2$).

Recall that $d_{\min}$ and $d_{\max}$ are respectively the minimum and maximum non-zero pairwise distance between points in $P$. It is not hard to see that the optimum solution to the instance has cost either 0 or at least $d_{\min}^{\ell}/\ell$. For a technical reason, we can redefine $d(p, q)$ as follows for every $p, q \in \mathbb{R}^D$:

$$d(p, q) = \begin{cases} 0 & \text{if } \|p - q\|_2 = 0 \\ \min\left\{\max\{\|p - q\|, \epsilon d_{\min}/(2n)\}, 2d_{\max}\right\} & \text{otherwise} \end{cases}.$$

That is, we truncate distances below at $\epsilon d_{\min}/(2n)$, and above at $2d_{\max}$. It is easy to see that the problem w.r.t the new metric is equivalent to the original one up to a multiplicative factor of $1 + \epsilon$. In the new instance, we have either $d(p, q) = 0$ or $d(p, q) \in [\epsilon d_{\min}/(2n), 2d_{\max}]$.

Given an integer $z' \in [0, n)$ and a set $C$ of $k$ centers, we define

$$\mathsf{cost}_{z'}(C) := \min_{P' \subseteq P : |P'| = n - z'} \sum_{p \in P'} d^{\ell}(p, C)$$

to be the cost of the solution $C$ to the $(k, z)$-median/mean instance defined by $P, d$ and $z'$. In the above definition, we remove $z'$ outliers and consider the cost incurred by the $n - z'$ non-outliers. Notice the set $P'$ that minimizes the cost is the set of $n - z'$ points in $P$ that are closest to $C$.

For some technical reason, we need to allow $z'$ to take real values in $[0, n)$. In this case, we define

$$\mathsf{cost}_{z'}(C) := \min_{w' \in [0,1]^P : w'(P) = n - z'} \sum_{p \in P} w'_p d^{\ell}(p, C).$$

Given a set $C$ of $k$ centers, the optimum $w'$ can be obtained in a greedy manner: assign 1 to the $n - \lceil z' \rceil$ points in $P$ that are closest to $C$, assign $\lceil z' \rceil - z'$ to the point in $P$ that is the $n - \lceil z' \rceil + 1$-th closest to $C$, and assign 0 to the remaining points.

### C.1  The $(k, z)$-Median/Means Problem Reformulated

In this section, we reformulate the $(k, z)$-median/means problems in a way that will be useful for our algorithm design. Given a threshold $L \geq 0$, we define $d_L(p, q) = \min\{d(p, q), L\}$ for every two

points $p, q \in \mathbb{R}^D$. In other words, $d_L$ is the metric $d$ with distances truncated at $L$. The following crucial lemma gives the reformulations of $k$-median/means problems:

**Lemma C.1.** *For any real number $z' \in [0, n)$, and any set $C$ of $k$ centers, we have*

$$\mathsf{cost}_{z'}(C) = \sup_{L \geq 0} \Big( \sum_{p \in P} d_L^\ell(p, C) - z' L^\ell \Big). \tag{1}$$

*Moreover, the superior is achieved when $L$ is the $(n - \lfloor z' \rfloor)$-th smallest number in the multi-set $\{d(p, C) : p \in P\}$.*

*Proof.* Let $\bar{L}$ be the $(n - \lfloor z' \rfloor)$-th smallest number in the multi-set $\{d(p, C) : p \in P\}$. Then it can be seen that $\mathsf{cost}_{z'}(C) = \sum_{p \in P} d_{\bar{L}}^\ell(p, C) - z' \bar{L}^\ell$. Indeed, $\mathsf{cost}_{z'}(C)$ is the sum of the $n - z'$ smallest numbers in $S := \{d^\ell(p, C) : p \in P\}$. (When $n - z'$ is not an integer, then we take a fraction of the last number.) To compute the quantity on the right side, we truncate the numbers in $S$ at $\bar{L}^\ell$, and then take the sum of the truncated numbers minus $z' \bar{L}^\ell$. Since $\bar{L}^\ell$ is the $(n - \lfloor z' \rfloor)$-th smallest number in $S$, this quantity is exactly $\mathsf{cost}_{z'}(C)$.

It remains to prove that $\sum_{p \in P} d_L^\ell(p, C) - z' L^\ell$ attains its maximum value at $L = \bar{L}$. First consider any $L < \bar{L}$, and define $P' = \{p \in P | L < d(p, C) < \bar{L}\}$, and $P'' = \{p \in P : d(p, C) \geq \bar{L}\}$. By the definition of $\bar{L}$, we have $|P''| \geq \lfloor z' \rfloor + 1 > z'$. Then, we have

$$\left( \sum_{p \in P} d_{\bar{L}}^\ell(p, C) - z' \bar{L}^\ell \right) - \left( \sum_{p \in P} d_L^\ell(p, C) - z' L^\ell \right)$$

$$= \sum_{p \in P'} (d^\ell(p, C) - L^\ell) + |P''|(\bar{L}^\ell - L^\ell) - z'(\bar{L}^\ell - L^\ell) \geq \sum_{p \in P'} (d^\ell(p, C) - L^\ell) \geq 0.$$

Now consider any $L > \bar{L}$ and define $P' = \{p \in P : \bar{L} < d(p, C) < L\}$ and $P'' = \{p \in P : d(p, C) \geq L\}$. By the definition of $\bar{L}$, we have $|P' \cup P''| = \big|\{p \in P : d(p, C) > \bar{L}\}\big| \leq \lfloor z' \rfloor \leq z'$. Then, we have

$$\left( \sum_{p \in P} d_{\bar{L}^\ell}(p, C) - z' \bar{L}^\ell \right) - \left( \sum_{p \in P} d_L^\ell(p, C) - z' L^\ell \right)$$

$$= - \sum_{p \in P'} (d^\ell(p, C) - \bar{L}^\ell) - |P''|(L^\ell - \bar{L}^\ell) + z'(L^\ell - \bar{L}^\ell)$$

$$\geq -|P'|(L^\ell - \bar{L}^\ell) - |P''|(L^\ell - \bar{L}^\ell) + z'(L^\ell - \bar{L}^\ell) \geq 0.$$

This finishes the proof of the lemma. $\qquad\square$

With the above lemma, the $(k, z)$-median/means problem becomes finding a set of $k$ centers $C \subseteq \mathbb{R}^D$ so as to minimize $\sup_{L \geq 0} \big( \sum_{p \in P} d_L^\ell(p, C) - z L^\ell \big)$. To get a handle on the problem, we first discretize the value space for $L$. Formally, we only allow $L$ to take values in

$$\mathbb{L} := \{0\} \cup \Big( \big\{ (1 + \epsilon)^t : t \in \mathbb{Z} \big\} \cap (\epsilon d_{\min}/(2(1 + \epsilon)n), 2 d_{\max}] \Big).$$

Then, we have $|\mathbb{L}| = O\left( \frac{\log(\Delta n / \epsilon)}{\epsilon} \right)$. We define $\mathsf{cost}_{z'}'(C)$ as in (1), except that we only consider $L$ values in $\mathbb{L}$. That is, for every $z' \in [0, n)$ and a set $C$ of $k$ centers, we define

$$\mathsf{cost}_{z'}'(C) := \sup_{L \in \mathbb{L}} \left( \sum_{p \in P} d_L^\ell(p, C) - z' L^\ell \right). \tag{2}$$

For a fixed $z'$ and $C$, we have $\mathsf{cost}_{z'}'(C) \leq \mathsf{cost}_{z'}(C)$, since the supreme is taken over a subset of $L$ values in the definition of $\mathsf{cost}_{z'}'(C)$. Now we show the other direction of the inequality:

**Lemma C.2.** *For every set $C$ of $k$ centers, and any $z' \in [0, n]$, we have*

$$\mathsf{cost}_{(1+\epsilon)^\ell z'}(C) \leq (1 + \epsilon)^\ell \mathsf{cost}_{z'}'(C). \tag{3}$$

*Proof.* By Lemma C.1, we have that $\mathrm{cost}_{(1+\epsilon)^\ell z'}(C) = \sup_{L \geq 0} \left( \sum_{p \in P} d_L^\ell(p, C) - (1+\epsilon)^\ell z' L^\ell \right)$. Let $\bar{L}$ be the $L \in \mathbb{R}$ that achieves the maximum value. Thus, $\mathrm{cost}_{(1+\epsilon)^\ell z'}(C) = \sum_{p \in P} d_{\bar{L}}^\ell(p, C) - (1+\epsilon)^\ell z' \bar{L}^\ell$. By Lemma C.1 and the new definition of the metric $d$, we have $\bar{L} = 0$ or $\bar{L} \in [\epsilon d_{\min}/(2n), 2d_{\max}]$. Thus there is always a $L' \in \mathbb{L}$ such that $\bar{L} \in [L', (1+\epsilon)L']$.

$$\mathrm{cost}_{(1+\epsilon)^\ell z'}(C) = \sum_{p \in P} d_{\bar{L}}^\ell(p, C) - (1+\epsilon)^\ell z' \bar{L}^\ell$$

$$\leq (1+\epsilon)^\ell \sum_{p \in P} d_{L'}^\ell(p, C) - (1+\epsilon)^\ell z' L'^\ell \leq (1+\epsilon)^\ell \mathrm{cost}'_{z'}(C).$$

The first inequality is by $L' \leq \bar{L} < (1+\epsilon)L'$ and the second inequality is by the definition of $\mathrm{cost}'_{z'}(C)$ and the fact that $L' \in \mathbb{L}$. $\qquad\square$

The lemma allows us to focus on the new objective function $\mathrm{cost}'_{\tilde{z}}(C)$ for some suitably defined $\tilde{z}$.

### C.2 Distributed Algorithm for the Reformulated Problem via $\epsilon$-Coresets

An important notion that has been used to design efficient algorithms for $k$-median/means in Euclidean space is the $\epsilon$-coreset. Roughly speaking, it is a weighted set of points that approximates the given set $P$ well. Formally,

**Definition C.3.** *A weighted set $(Q, w)$ of points is an $\epsilon$-coreset for $P'$ w.r.t. distance $d'$, if for every set $C \subseteq \mathbb{R}^D$ of $k$ centers, we have*

$$\left( \sum_{q \in Q} w_q d'^\ell(q, C) \right) \Big/ \left( \sum_{p \in P'} d'^\ell(p, C) \right) \in [1 - \epsilon, 1 + \epsilon].$$

The following theorem from [2] gives a distributed algorithm to construct $\epsilon$-coresets for the points $P$ and a truncated metric $d_L$:

**Theorem C.4.** *[2] Given $\delta > 0, \epsilon > 0, L \geq 0$, there is an 2-round distributed algorithm that outputs an $\epsilon$-coreset $(Q, w)$ of $P$ w.r.t distance $d^L$, with probability at least $1 - \delta$. The size of the coreset is at most $\Phi$, where $\Phi = O\left( \frac{1}{\epsilon^2}(kD + \log \frac{1}{\delta}) + mk \right)$ for $k$-median, and $\Phi = O\left( \frac{1}{\epsilon^4}(kD + \log \frac{1}{\delta}) + mk \log \frac{mk}{\delta} \right)$ for $k$-means. The communication complexity of the algorithm is $O(D\Phi)$.*

The correspondent theorem in [2] only considers the original Euclidean metric $\| \cdot - \cdot \|_2$. In our definition of $d_L$, we truncated distances below at $\epsilon \cdot d_{\min}/(2n)$, and then above at $L$. But it is easy to extend their theorem so that it works for the truncated metrics, since all we need is that the metric has $O(D)$ "pseudo-dimension" (defined in [2]). Truncating the metric only change the pseudo-dimension by an additive constant. From now on, let $\Phi$ be the upper bound on the size of the $\epsilon$-coreset in Theorem C.4.

With Theorem C.4 in hand, it is straightforward to give our algorithm for $(k, z)$-median/means. For all $L \in \mathbb{L}$, we run in parallel the 2-round distributed algorithm in Theorem C.4 with $\delta$ scaled down by a factor of $|\mathbb{L}|$ to obtain a $\epsilon$-coreset $(Q_L, w_L)$. The communication cost of the algorithm is then $\Phi D \cdot \frac{\log(n\Delta/\epsilon)}{\epsilon}$.

Let $\tilde{z} = \frac{(1+\epsilon)^2 z}{1-\epsilon}$. We would like to find a set $\tilde{C}$ of $k$ points that minimizes $\sup_{L \in \mathbb{L}} \left( \frac{1}{1-\epsilon} \sum_{q \in Q_L} w_q d_L^\ell(q, \tilde{C}) - \tilde{z} L^\ell \right)$. However, it is not even clear whether the optimum $\tilde{C}$ can be represented using finite number of bits or not. Instead, the coordinator will output a set $\tilde{C} \subseteq \mathbb{R}^D$ of $k$ centers such that for every set $C^* \subseteq \mathbb{R}^D$ of $k$ centers, we have

$$\sup_{L \in \mathbb{L}} \left( \frac{1}{1-\epsilon} \sum_{q \in Q_L} w_q d_L^\ell(q, \tilde{C}) - \tilde{z} L^\ell \right) \leq \sup_{L \in \mathbb{L}} \left( \frac{1+\epsilon}{1-\epsilon} \sum_{q \in Q_L} w_q d_L^\ell(q, C^*) - \tilde{z} L^\ell \right). \tag{4}$$

The extra $(1 + \epsilon)$ factor on the right-side allows us to partition the Euclidean space into finite number of cells. This can be done by partitioning the space into $O\left(\frac{\log(\Delta n/\epsilon)}{\epsilon}\right)^{|\mathbb{L}|\Phi}$ cells so that all points in a cell have similar respective distances to all points in $\bigcup_{L \in \mathbb{L}} Q_L$. So, we can choose an arbitrary representative point from each cell, and then enumerate all sets $\tilde{C}$ of $k$ representatives and output the one with the minimum $\sup_{L \in \mathbb{L}} \left(\frac{1}{1-\epsilon} \sum_{q \in Q_L} w_q d_L^\ell(q, \tilde{C}) - \tilde{z}L^\ell\right)$. The running time of the algorithm can be bounded by $\exp\left(\Phi, k, |\mathbb{L}|, D, \log\left(\frac{\log(n\Delta/\epsilon)}{\epsilon}\right)\right) = \exp\left(\text{poly}\left(\frac{1}{\epsilon}, k, D, m, \log \frac{1}{\delta}, \log \Delta\right)\right)$.

### C.3  Analysis of the algorithm

We now show that the algorithm gives a $(1 + O(\epsilon), 1 + O(\epsilon))$-approximation algorithm to the $(k, z)$-median/means problem. With probability at least $1 - \delta$, for every $L$, the weighted set $(Q_L, w_L)$ is an $\epsilon$-corset for $P$ w.r.t metric $d_L$. Let $C^*$ be the optimal set of centers for the original $(k, z)$-median/means problem. Then, for every $z' \in [0, n]$, we have

$$
\begin{aligned}
&\text{cost}'_{\tilde{z}}(\tilde{C}) \\
&= \sup_{L \in \mathbb{L}} \left(\sum_{p \in P} d_L^\ell(p, \tilde{C}) - \tilde{z}L^\ell\right) \qquad\qquad \leq \sup_{L \in \mathbb{L}} \left(\frac{1}{1-\epsilon} \sum_{q \in Q_L} w_q d_L^\ell(q, \tilde{C}) - \tilde{z}L^\ell\right) \\
&\leq \sup_{L \in \mathbb{L}} \left(\frac{1+\epsilon}{1-\epsilon} \sum_{q \in Q_L} w_q d_L^\ell(q, C^*) - \tilde{z}L^\ell\right) \quad \leq \sup_{L \in \mathbb{L}} \left(\frac{(1+\epsilon)^2}{1-\epsilon} \sum_{p \in P} d_L^\ell(p, C^*) - \tilde{z}L^\ell\right) \\
&= \frac{(1+\epsilon)^2}{1-\epsilon} \sup_{L \in \mathbb{L}} \left(\sum_{p \in P} d_L^\ell(p, C^*) - zL^\ell\right) \quad = \frac{(1+\epsilon)^2}{1-\epsilon} \text{cost}'_z(C^*).
\end{aligned}
$$

The first and the third inequalities are by the definition of $\epsilon$-coreset, while the second inequality is by (4). Then with Lemma C.2, we know that

$$
\begin{aligned}
\text{cost}_{\frac{(1+\epsilon)^{\ell+2}}{1-\epsilon}z}(\tilde{C}) = \text{cost}_{(1+\epsilon)^\ell \tilde{z}}(\tilde{C}) &\leq (1+\epsilon)^\ell \text{cost}'_{\tilde{z}}(\tilde{C}) \\
&\leq \frac{(1+\epsilon)^{\ell+2}}{1-\epsilon} \text{cost}'_z(C^*) \leq \frac{(1+\epsilon)^{\ell+2}}{1-\epsilon} \text{cost}_z(C^*).
\end{aligned}
$$

So, $\tilde{C}$ is a $\left(\frac{(1+\epsilon)^{\ell+2}}{1-\epsilon}, \frac{(1+\epsilon)^{\ell+2}}{1-\epsilon}\right) = (1 + O(\epsilon), 1 + O(\epsilon))$-approximate solution. We can scale down the input $\epsilon$ by a constant factor to obtain a $(1 + \epsilon, 1 + \epsilon)$-approximation.

As we mentioned, the running time of the algorithm for the central coordinator is exponential in $\frac{1}{\epsilon}, k, D, m, \log \frac{1}{\delta}$ and $\log \Delta$. For each machine $i$, the running time in the algorithm of [2] is dominated by the time to compute an $O(1)$-approximation for the $k$-median/$k$-means problem for $P_i$, which is polynomial in $n_i$ and $D$.

## D  Complete Experiment Results

### D.1  $k$-Center Clustering with Outliers

We evaluate the performance of our $(k, z)$-center algorithm (Algorithm 3) on several real-world datasets, which are summarized in Table 1. In the experiments we compare dist-kzc with many other $k$-center methods, including two centralized methods (greedy [9] and kzc [3] and four distributed methods (random-random, random-kzc, MKCWM [12], and GLZ [6]). The greedy method has a 2-approximation ratio in the no-outlier scenario, but doesn't take outliers into account. The random-random and random-kzc methods serves as two baselines: random-random randomly sample $k + z$ points on each machine, then further randomly choose $k$ points from the total $m(k + z)$ sampled points as final cluster centers; random-kzc is similar to random-random, except that it chooses the final $k$ centers by the kzc method. The MKCWM and GLZ are the state-of-art distributed $k$-center algorithms that handle outliers. For each parameter setting the experiment is repeated for 5 runs and

the average result is reported. Note the three distributed baseline methods random-random, random-kzc, and MKCWM all have the same communication cost $md(k + z)$, while GLZ's communication cost is $\tilde{O}(mk + m/\epsilon)$. All methods are implemented in Python and the experiments are conducted on a 2-core 2.7 GHz Intel Core i5 laptop.

| Name | Size: $n$ | Dimension: $B$ |
|---:|---|---|
| spambase[2] | 4,601 | 57 |
| parkinsons[3] | 5,875 | 16 |
| pendigits[2] | 10,992 | 16 |
| letter[2] | 20,000 | 16 |
| skin[2] | 245,057 | 3 |
| covertype[2] | 581,012 | 10 |
| gas[2] | 928,991 | 10 |
| power[2] | 2,049,280 | 7 |

Table 1: Clustering datasets used for evaluation

The experiments consist of two parts: In the first part we compare our algorithms with the two centralized methods. This part is conducted only for the 4 smaller datasets (*spambase*, *parkinsons*, *pendigits*, and *letter*), on which centralized methods can finish in an acceptable time. In the second part we compare our algorithms with other distributed methods on the 4 larger datasets (*skin*, *covertype*, *gas*, and *power*).

**Distributed v.s. Centralized:** Figure 1 and Figure 2 show the results on the four smaller datasets. Figure 1 demonstrates how the objective value and communication cost change with $z$ when $k$ is fixed to 20. Our algorithm dist-kzc always achieve comparable objective with other distributed baselines. On datasets *spambase* and *parkinsons*, the objective value even matches the best centralized method (kzc). When it comes to communication cost, dist-kzc shows a clear advantage over random-random, random-kzc, and MKCWM, which matches our theoretical results.

Figure 2 depicts the performance with respect to different value of $k$ when $z$ is fixed to 256. dist-kzc still achieves similar (or better) objective values among all distributed methods. But we can see that when $k$ increases, the communication cost of dist-kzc ($\epsilon = 0.1$) approaches those of other distributed methods. Recall that the communication cost of dist-kzc is $\tilde{O}(mk/\epsilon)$ which can be similar to $O(m(k + z))$ when $z$ and $k/\epsilon$ are in the same order. If we choose a large value of $\epsilon = 0.99$, the communication cost of dist-kzc becomes much stable, while the objective value is only slightly worse. This suggests that in practice we can choose a relatively large $\epsilon$ to obtain small communication cost.

We want to remind the readers that the approximation ratio of dist-kzc holds for removing $(1 + \epsilon)z$ outliers, while in the experiments the objective is computed by removing only $z$ outliers. This indicates that dist-kzc may have better performance than what is predicted theoretically.

**Large scale:** This part contains experiment results on the four large datasets: *skin*, *covertype*, *gas*, and *power*. The GLZ method needs solving many local $(k, z')$-center instances, which is too slow to finish on these large datasets. Hence here we use its variant provided by [6], denoted as GLZ-z. GLZ-z works similar as GLZ, but avoids solving $(k, z')$-center locally on each machine by transmitting $\tilde{O}(mk + z)$ data to the coordinator. So GLZ-z has a higher communication cost than GLZ, but it's still much better than MKCWM which has a $O(m(k + z))$ communication cost.

Similar to the previous part, Figure 3 and Figure 4 show results for varying $z$ and $k$. Our method still achieves comparable objective value with the best distributed baselines. The communication cost of our algorithm is always much smaller than MKCWM, and matches that of GLZ-z. This advantage is more obvious with bigger $z$, but here to make all the baselines terminate in acceptable times we only use $z \sim \sqrt{n}$.

Figure 1: Centralized vs. Distributed, with varying $z$ and fixed $k = 20, m = 5$.

Figure 2: Centralized vs. Distributed, with varying $k$ and fixed $z = 256, m = 5$.

Figure 3: Large scale, with varying $z$

Figure 4: Large scale, with varying $k$

## D.2 $k$-Means Clustering with Outliers

**The centralized solver:** We test our distributed $k$-means algorithm proposed in section C. As we described, the algorithm requires solving a *min-max* clustering problem on the coordinator. Formally, given a set of datasets $Q_1, \ldots, Q_M$, each equipped with its own metric $d_1, \ldots, d_M$, the goal is to find a center set $C$ minimizing the maximum cost over all $M$ datasets:

$$\min_{C:|C|=k} \max_{i \in [M]} \sum_{p \in Q_i} d_i^l(p, Q_i) \tag{5}$$

where $l = 1, 2$ corresponding to the $k$-median or $k$-means objective respectively.

Although we don't know any practical algorithm for such min-max clustering problem, there exists some results addressing a simpler form of the min-max $k$-median problem: Suppose there're only $N$ possible locations for selecting the center set $F$ (i.e., $C \subset V$ for some $|V| = N$), and every dataset $Q_i$ has the same metric $d$, then Anthony *et al.* [1] shows that a simple reverse-greedy method achieves $O(\log N + \log M)$-approximation for the min-max $k$-median problem in this special case. We adapt their method to solve our min-max $k$-means problem in the experiment. For completeness, the algorithm is listed below:

---

**Algorithm D.1** reverse-greedy $(k, \{(Q_i, d_i)\}_{i=1}^M, B)$[1]

---

1: $C^1 \leftarrow \bigcup_{i=1}^M Q_i, w_i^1 \longleftarrow 1$ for all $i \in [M]$;
2: **for** $t \leftarrow 1$ to $N - k$ **do**
3:     For every $v \in F^t$ and $i \in [M]$, let $\delta_i^t(v) \leftarrow \sum_{p \in Q_i}(d_i^2(p, C^t \setminus \{v\}) - d_i^2(p, C^t))$
4:     $\hat{C}^t \leftarrow \{v \in C^t | \forall i \in [M], \delta_i^t(v) \leq B/2\}$
5:     $v^t \leftarrow \arg\min_{v \in \hat{C}^t} \sum_{i=1}^M w_i^t \cdot \delta_i^t(v)$
6:     For all $i \in [M]$, let $w_i^{t+1} \leftarrow w_i^t \left(1 + \frac{1}{B}\right)^{\delta_i^t(v^t)}$
7: **return** $C^{N-k+1}$

---

Roughly speaking, the algorithm starts with $C$ being the set of all points, and iteratively remove points in $C$ until it shrinks to size $k$. In each iteration the algorithm removes from $C$ the point that incurs the least weighted total cost increase. However, because our problem is more general than that in [1], we don't know whether their approximation guarantee for Algorithm D.1 still holds here.

**Algorithms:** We compare our implementation with some other algorithms for the $k$-means/$(k, z)$-means problem, including two centralized ones and two distributed ones: k-means [11], the classical Lloyd's algorithm; k-means$--$ [4], like k-means, but uses some heuristics to handle outliers; BEL [2], the distributed $k$-means algorithm based on coreset; and CAZ [5], a recently proposed

distributed $(k, z)$-means algorithm. The BEL and CAZ algorithms both belong to the two-level clustering framework[7]: first construct a local summary on each machine and aggregate them on the coordinator, then the coordinator conduct a centralized clustering over the aggregated summaries to get the final result. But the main focus of BEL and CAZ is how to construct local summary, and they don't specify the actual coordinator solver used. In the experiment we use k-means and k-means−− as the centralized solver for BEL and CAZ respectively. All methods are implemented in Python and the experiments are conducted on a 2-core 2.7 GHz Intel Core i5 laptop.

**Datasets:** The experiment is conducted on one synthesized dataset and three real-world datasets. The real-world datasets are *spambase, parkinsons*, and *pendigits* (see Table 1). Unlike the $k$-center case, the outliers in the original dataset are unable to significantly affect the objective value. Thus to make the algorithm's effect clearer, we manually add 500 outlier points to each of the three dataset. The synthesized dataset is sampled from a mixture of Gaussian model, of which the parameters are also randomly generated; specifically, we sample 10000 points in total from 4 different Gaussian distributions in $\mathbb{R}^5$, and manually add another 500 outliers to the dataset.

**Parameter setting:** For each dataset, we fix $k$ and vary $z$. On the three real-world datasets, $k$ is set to be 10 and $z$ varies from $2^5$ to $2^{11}$; on the synthesized dataset, $k$ is set to be 4 and $z$ ranges from $2^6$ to $2^{10}$. The number of machines are fixed to 5 for all 4 datasets. Throughout the experiment, we use $\epsilon = 0.3$ as the error parameter for our algorithm. We measure how the objective value and communication cost (for distributed methods only) changes with $z$. But different from the setting in Section D.1, here we compute the cost of our method by removing $(1 + \epsilon)z$ outliers to match our theory result. (In this sense, the comparison is "more fair" for us than in Section D.1)

Another issue in applying our $(k, z)$-means algorithm is the choice of appropriate coreset size. Unlike the result for our $(k, z)$-center algorithm, we only have an asymptotic estimation for the coreset size, which is not so instructive in practice. Therefore, in the experiment we hand-pick the coreset size by some heuristics: when the value of the error parameter $\epsilon$ is given, we can compute the total number of different threshold distance that will be tried (i.e., $|\mathbb{L}|$). Then we choose the coreset size to be $\max\left\{10k, \frac{n}{10m|\mathbb{L}|}\right\}$. So each coreset contains at least $10k$ samples, and when $n \gg km|\mathbb{L}|$, we allow the total size to be as large as $n/10$.

Figure 5: Comparison of the our distributed $(k, z)$-means implementation with other distributed/centralized methods. The first row is objective value, and the second is communication cost.

**Experiment results:** Figure 5 shows the experiment result: we can see that our algorithm performs surprisingly well in terms of objective value, often achieving the lowest cost among all the methods. The effect of outliers is most clearly revealed on the synthesized data, where BEL and k-means perform significantly worse than others. In particular, although we remove $\epsilon z$ more outliers when calculating the cost for our method, it's still much better than BEL even if compared at different $z$: consider our method's cost at $z = (1 + \epsilon)2^7 = 1.3 \cdot 2^7$ with BEL's at $z = 2^8$.

The communication cost of our method doesn't change with $z$, since the way we decide the coreset size makes it fixed. BEL's communication cost is also not affected by $z$, as it doesn't deal with outliers. In contrast, CAZ's communication is in the order of $O(mk \log n + z)$, which is reflected in the figure as it grows linearly in $z$. Although our centralized solver uses some heuristics and thus doesn't have provable guarantees, the experiment results suggest that our coresets construction indeed preserves the outliers information while being independent of $z$.

## Footnotes

[1] This is a well-known result in communication complexity theory, see e.g. [8]

[2] The UCI data repository [10]

[3] [13]