[Reviews · NeurIPS 2018]

Reviewer 1



The paper describes a distributed bicriteria approximation algorithm for k-center with z outliers. It uses 24(1+eps)k centers and (1+eps)z outliers. Its communication cost does not depend on z. They show that communication cost of z is unavoidable if you insists on having only z outliers with a simple example on the line. In the suggested protocol each machine first locally aggregates its data and sends the aggregated data to a coordinator that computes the clustering using the aggregated data with a known centralized algorithm. A similar scheme is suggested for k-means and k-median that uses coresets, but the centralized computation time is exponential. The algorithms are simple. minor comments: -- line 58: what do you mean by solving the problem COMPLETELY ? -- line 59: I think you did not say that m is the number of machines -- line 199: Why is there an "s" in the equation ? -- line 294: "need output" ==> "need to output" -- Why do you need so many primes in the notation of Theorem 2.1 -- line 232: seems that w'(q) is just a definition you need and not part of the claim

Reviewer 2



Clustering large datasets with outliers is a fundamental problem. In the distributed setting, it becomes quite challenging and only recently [1,2], there has been a substantial progress on the upper and lower bounds on the approximation ratio and communication complexity. The present paper essentially resolves both questions in the case when there are a known number/fraction of outliers. This is a major advance, clearly worth acceptance to NIPS. The discussion of the related work omits [2], which it should not. In particular, the lower bound for the blackboard model is features an O(m + k) term, for m machines and k cluster, rather than O(mk) or O(mk log mk) in the upper bound by the present authors. Some table or discussion would be very welcome. There is another NIPS submission on the same problem [3]. It would be nice if the authors could comment on the relative practicality (if not present an empirical comparison). UPDATED AFTER REBUTTA: I appreciate the authors promise to do the empirical comparison. [1] https://dl.acm.org/citation.cfm?id=3087568 [2] http://papers.nips.cc/paper/6560-communication-optimal-distributed-clustering [3] https://arxiv.org/abs/1805.09495

Reviewer 3



The paper describes algorithms for distributed k-center/k-median/k-means clustering. For distributed k-center, it is known that any constant approximation needs Omega(z) communication cost, where z is the number of allowed outliers. This can be avoided by using a bicriteria approximation where more than z outliers are allowed, but the cost is compared to the best solution with at most z outliers. It is known that using 2z outliers is sufficient to obtain a O(1)-approximation. The paper at hand now improves the violation with respect to the number of outliers to (1+eps)*z. The result is given for metric k-center, but works for Euclidean k-center as well (since Euclidean distances are a metric, and only O(1) is obtained anyway). For Euclidean k-median/k-means, a similar result is obtained using coresets. However, here the interpretation is a bit different, since these problems can (under no assumptions on n and d) not be (1+eps)-approximated. (This still needs to be done by the coordinator after collecting the information). Nevertheless, coresets have proven useful for plain k-median/k-means as well, so I consider this extension to be meaningful. The paper is well written and contains a helpful figure for understanding the main proof. The algorithm is elegant. The experiments have not made it into the actual paper, which I think is not so good. The experiments do look solid and compare several approaches. I hope that a long version of this paper is going to be published soon after the publication of the conference version so that this part of the paper is not lost to the reader. The one-sentence summary of it in the actual conference version is not very informative. The overall accessment probably hinges on the question how important the improvement from 2z to (1+eps)z is. - The sentence on [5] in line 114 should have k-median in it and not k-center. - To the conclusion: It seems unlikely that the problem can be solved within a 1+eps factor on the coreset for k-median/k-means, since the problems are even APX-hard without the outlier part. Edit after rebuttal: Since I cannot pinpoint an exact minus point for this paper and the result is interesting, I've update my score to accept.